# Glial Response and Neuroinflammation in Cerebrocortical Atrophy in a Young Irish Wolfhound Dog

**DOI:** 10.3390/ani11010143

**Published:** 2021-01-11

**Authors:** Fabiano J. F. de Sant’Ana, Miguel Omaña, Ester Blasco, Martí Pumarola

**Affiliations:** 1Laboratório de Diagnóstico Patológico Veterinário, Universidade de Brasília, Brasília 70636-020, Brazil; 2Hospital Veterinario Canis/MRIVETS, 07101 Palma de Mallorca, Spain; miguelomana@yahoo.es; 3Unit of Murine and Compared Pathology, Faculty of Veterinary, Universitat Autònoma de Barcelona, 08193 Bellaterra, Spain; ester.blasco@uab.cat (E.B.); marti.pumarola@uab.cat (M.P.); 4Biomaterials and Nanomedicine (CIBER-BBN), Networking Research Center on Bioengineering, Universitat Autònoma de Barcelona, 08193 Bellaterra, Spain

**Keywords:** cerebral cortical atrophy, immunohistochemistry, neuronal necrosis, neuropathology

## Abstract

**Simple Summary:**

Neuroinflammation is considered a reaction of the nervous system itself to protect and repair structural changes developed in it. Despite its initial positive purpose, sometimes it can produce worse consequences for the tissue. In this article, we present a case of an Irish Wolfhound dog suffering a rare idiopathic neurodegenerative disease producing wide cerebral cortical damage with loss of neuronal bodies in a bilateral and symmetrical pattern. We have studied the glial components of the neuroinflammation developed describing how they have exacerbated the nervous tissue damage.

**Abstract:**

A two-year-old, Irish Wolfhound dog presented with a history of progressive neurological signs. Neurological exam revealed disorientation, absence of menace response, reduction of right nasal sensation, hypermetria and ataxia with reduction of proprioception in all four limbs. MRI findings were compatible with laminar neuronal necrosis and possible bilateral cortical cerebral atrophy. Grossly, a severe bilateral reduction of the gray matter with flattening of gyri, mainly in frontal and parietal cerebral areas, was observed. Histologically, multiple, segmental, bilateral, and symmetric areas of neuronal loss, necrosis and degeneration, in a laminar pattern, associated with a reactive gliosis were observed. Immunohistochemical studies showed severe reduction of neuronal bodies, proliferation and hypertrophy of astrocytes and microglia. Few perivascular B and T cells were demonstrated. Based on these data, we show some of the neuroinflammatory events that occur during CNS repair in a chronic phase of this condition.

## 1. Introduction

Degenerative neurological diseases in domestic animals include a broad group of disorders that are characterized by progressive, bilateral and symmetrical degeneration and loss of cells, mainly neurons. Most of these diseases are of a genetic basis, but the precise pathogenic mechanisms are still poorly known or understood. Breed predisposition seems to occur in some cases. These diseases that affect the central nervous system can be classified in neuronal degenerations, axonal degenerations, myelin disorders, storage diseases, spongiform encephalopathies, spongy degenerations, and selective symmetrical encephalomalacias [1]. Recently, a novel idiopathic condition, characterized by superficial neocortical degeneration was reported in five dogs from North America and United Kingdom [2].

Neuroinflammation, the activation of the neuroimmune cells (microglia and astrocytes) into proinflammatory states, with no known causative insult and little change in blood-brain barrier biology, has been suggested as a pathological contributor in several neurodevelopmental, psychiatric, and neurodegenerative disorders [3]. This sustained inflammatory response suggests an important role of effectors of neuroinflammation in neuronal dysfunction and death [4]. Neuroinflammation is usually referred to as the chronic response of nervous tissue [3]. Neuroinflammation was previously understood as a local tissue response with few or no involvement of the peripheral immune system. Nevertheless, recent data support it is influenced by a number of peripheral and other factors, such as cytokines [5], chemokines expressed by lymphocytes [6], plasma kinins [7], hormones, and a complex of molecular interactions [4].

Here, we describe the pathological and immunohistochemical findings in one case of cerebrocortical atrophy in a young Irish Wolfhound dog, with emphasis on the glial reaction (neuroinflammation) and on the poor lymphocytic response to neuronal injury, which has not been previously investigated.

## 2. Description of the Case

A two-year-old, Irish Wolfhound, male dog presented with one-month history of progressive neurological signs, including difficulty to jump, lumbar pain and pelvic limb weakness. Hematological and biochemical (blood urea nitrogen, creatinine, glucose, albumin, total proteins, alanine aminotransferase, alkaline phosphatase) analyses did not reveal changes. Radiography of thorax was normal. In the neurological exam, disorientation, absence of menace response, reduction of right nasal sensation, hypermetria and ataxia with reduction of proprioceptive positioning in all four limbs with normal spinal reflexes were observed. These findings were indicative of diffuse primary bilateral cortico-thalamic lesions, with perhaps more severe involvement of the right cortical region, as well as the cerebellum. Magnetic resonance imaging (MRI) showed hyperintensity and increased width of the subarachnoid space surrounding the cerebral gyri in T2 weighted images. With similar distribution, these regions presented hypointensity in T1 and FLAIR (Fluid attenuated inversion recovery) sequences, and the cerebrospinal fluid sign was isointense. A line of hyperintensity was observed in the neocortical region in the FLAIR sequence, suggesting laminar neuronal necrosis (Figure 1). After the administration of contrast, there were zones of contrast enhancement in the neocortex and the meninges. MRI findings demonstrated possible bilateral cortical cerebral atrophy. The analysis of cerebrospinal fluid collected from the cerebellomedullary cistern revealed neutrophilic pleocytosis (50 cells/µL, 82% polymorphonucleates and 18% lymphocytes). RT-PCR analysis to six neurologic infectious diseases of dogs (canine distemper, toxoplasmosis, neosporosis, borreliosis, bartonellosis, and cryptococcosis) were negative. The animal was euthanized due to the poor prognosis and decision of the owner. Necropsy was performed and the brain was submitted to histopathological examination.

After fixation in 10% neutral buffered formalin for four days, transverse sections of the brain were performed. Representative fragments of cerebrum (frontal, parietal, temporal, occipital, and piriform cortical areas, basal nuclei, and hippocampus), thalamus, midbrain, pons, cerebellum and medulla oblongata were processed for routine histopathological examination upon hematoxylin and eosin staining. In addition, immunohistochemical (IHC) evaluation was performed using a biotin-peroxidase system and diaminobenzidine as the chromogen. Antigen retrieval was performed with citrate buffer pH 6.0 (NeuN, GFAP, Iba1 and CD20) or 0.1% protease (CD3). To block the endogenous peroxidase activity, the slides were incubated in a solution of H_2_O_2_ (3%) in distilled water. The reagents were applied manually, with an over-night incubation at 4 °C for the monoclonal primary antibodies and a 40 (NeuN, GFAP, CD3 and CD20) to 60 min (Iba1) incubation for the secondary antibodies. An avidin-biotin complex solution was used in the case of Iba1 to amplify the response and was incubated for 1 h. The diaminobenzidine chromogen was applied for 10 min. The IHC antibody panel is described in Table 1. The IHC sections were counterstained using Harris hematoxylin. The positive controls for IHC consisted of brain (NeuN, GFAP, and Iba-1) and lymph node (CD3 and CD20) of a dog without morphologic changes. For the negative controls, an isotype-specific immunoglobulin was used as a substitute for the primary antibody and no immunostaining was detected in these sections.

Grossly, a severe reduction in neocortical gray matter with flattening of gyri bilaterally, affecting mainly frontal and parietal areas, was observed (Figure 2). In the surface of transverse sections, thinning of the frontal, parietal, and temporal cortices was more evident, and there was no clear distinction between gray and white matter. Gross changes in the occipital cortex, cerebellum, and brainstem were not observed.

Histologically, irregular, multiple, segmental areas of thinning of the gray matter with a bilateral and symmetrical pattern were observed in the frontal, parietal, temporal, and occipital cortices. Superficial layers (laminae I and II) were pallid due to severe absence of neuronal bodies and microspongiosis of neuropile (Figure 3A). In addition, individual neuronal necrosis and deposition of proteinaceous eosinophilic globules were observed. These changes were observed equally in the surface of the gyri and in deepest part of sulci. Numerous foamy macrophages (gitter cells) and a mild infiltration of lymphocytes was also noted, mainly in the subarachnoid space that was distended secondarily to cortical atrophy. In deepest layers (laminae IV and V), moderate depletion of neuronal bodies was associated with a reactive gliosis. Subcortical white matter showed disorganization and mild to moderate spongiosis. In the hippocampus, individual neuronal necrosis with reactive gliosis was observed in the CA2 and CA4 regions. Significant lesions were not detected in the other regions evaluated. The severe decrease in number of neuronal bodies in the affected cortex was confirmed by NeuN immunostaining. The few layers with remaining neurons were disorganized (Figure 3B). Glial Fibrillary Acidic Protein (GFAP) labelling showed evident proliferation and hypertrophy of astrocytes in the affected gray matter, mainly in the more superficial (adjacent to pia mater) and also in the deepest layers. Reactive astrocytes frequently presented enlarged nuclei and abundant and extended cytoplasmic processes (Figure 3C). Proliferation and hypertrophy of immunopositive Iba-1 microglia cells were also evident in the affected gray matter areas (Figure 3D). Hypertrophied microglia were observed mainly in the deeper neuronal layers and in the subarachnoid space. CD20 and CD3 immunolabeling demonstrated the presence of few perivascular B and T cells, respectively, located in the subarachnoid space and rarely in the subpial areas of the affected cortex.

## 3. Discussion

The current manuscript describes the neurohistological and immunohistochemical study of an unusual case of cerebrocortical chronic selective bilateral laminar neuronal degeneration and necrosis in a two-year-old dog. A recent retrospective study (1982–2012) described a very similar neuropathological condition in five young dogs of both sexes from United Kingdom and USA [2]. Three out of these five dogs were related hounds (two Irish Wolfhound and one Scottish Deerhound). In the present study, the affected dog was an Irish Wolfhound that lived in Spain, though it was initially acquired in Germany. These data indicate a strong probability of a hereditary and genetic basis for this disease, although the pathogenesis of the condition of insidious onset remains unknown. Few inherited disorders have been described and recognized in Irish Wolfhound dogs, such as dilated cardiomyopathy, hip dysplasia, and portosystemic shunts [8,9,10,11]. In addition, hyperekplexia (Startle disease), a rare severe congenital disorder, has been recognized as one of the few inherited neurological diseases diagnosed in this canine breed, usually in young puppies [7].

Clinical signs of this chronic neurodegenerative disorder of young dogs are progressive and include mainly ataxia, paresis, hyperextension of limbs, blindness, difficulty of prehension, seizures, and hyperesthesia presented over weeks to months [2]. Some of these signs were observed in the current case. Our MRI findings are in accordance to the ones reported previously [2]. Based on the imaging findings, we could not objectively determine cortical atrophy due to the limited number of studies on cortical thickness in dogs. Clinical history and neurological exam were suggestive of diffuse cerebral lesion affecting the neocortex (sensory and visual deficits) and brainstem (consciousness or proprioceptive deficits), though the MRI only suggested a severe neocortical degenerative and/or necrotic lesion. Although hypermetria was suggestive of cerebellar involvement, we could not find any structural change in this region to explain this clinical sign. The cerebellum is a structure commonly affected in neurodegenerative diseases of domestic animals, mainly in cerebellar cortical abiotrophies (1). There were no scientific evidences to explain the high selectivity of some neuronal groups in these diseases. Systemic infectious diseases or metastasis affecting the CNS can cause multifocal lesions and similar clinical signs. However, the negative results of PCR investigations performed in this study for six important systemic diseases of dogs and the image findings made less probable these potential causes.

The histopathological pattern of this condition is considered atypical, because usually in domestic animals with cerebrocortical degeneration and necrosis, several layers of neurons are injured [12,13]. Here, we observed a chronic and selective neuronal loss mainly affecting superficial (and eventually deeper) layers of the neocortex and hippocampus, as previously reported [2]. Occasionally, other regions of the brain, such as hypothalamus, mesencephalon, and pons can also be affected [2], but no changes were detected in our case. Our histopathological findings indicate an expressive response of astrocytes and microglial cells to the chronic severe neuronal damage. The proliferation and activation of both glial cells in this neurodegenerative condition were considered crucial to the repair, regulation, phagocytosis and structural support of the injured tissue. These cells provide pre- and anti-inflammatory actions in various functions under basal and disease conditions (3).

In the current study, immunohistochemical evaluation was effective to demonstrate the exclusive glial response to neuronal changes, characterized by proliferation and hypertrophy of astrocytes and microglia, particularly in the most affected areas of neocortex. These findings are associated with the chronic progressive loss and injury of cortical neurons highlighted by labelling with NeuN. In the current study, the lymphocyte-poor immune response to the neuronal damage was restricted to subarachnoid space and subpial areas. The findings of the current study suggest that an initial glial response produced to control and repair the neuronal damage was converted in an exaggerated response (neuroinflammation) and negative to the nervous system, with increased tissue injury. Future studies evaluating the complex neuroinflammatory pathways and genetic features of this uncommon neurodegenerative disorder are needed to further characterize its etiology and pathogenesis. The fact that inflammation can be either protective or damaging, is fundamental to understand the pathogenesis of brain disorders.

## 4. Conclusions

Based on histopathological and immunohistochemical results of the current study, we suggest that exclusive glial response, including proliferation and hypertrophy of astrocytes and microglia, is a crucial event in the chronic neuronal damage, observed in the selective superficial neocortical neuronal degeneration of Irish Wolfhound dogs, without effective involvement of the lymphocytic response.

## Figures and Tables

**Figure 1 animals-11-00143-f001:**
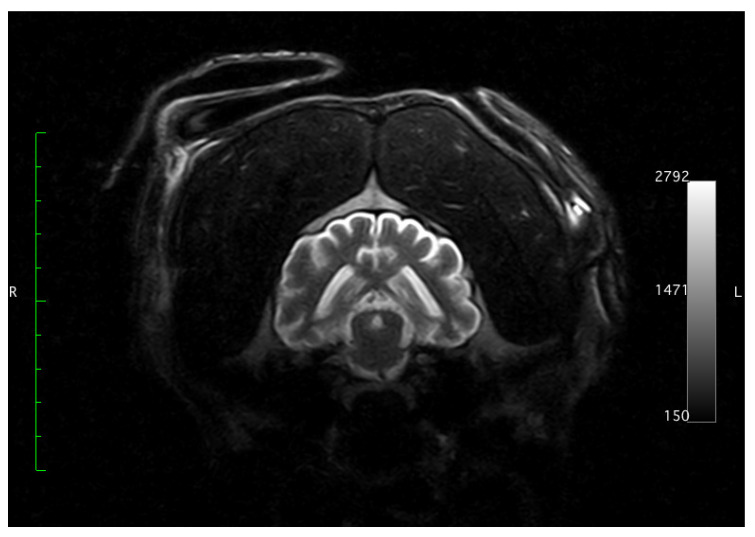
Brain magnetic resonance imaging from a young Irish Wolfhound dog with bilateral cerebrocortical atrophy. There is a line of hyperintensity in the neocortical region in the FLAIR transverse sequence, compatible with laminar neuronal necrosis.

**Figure 2 animals-11-00143-f002:**
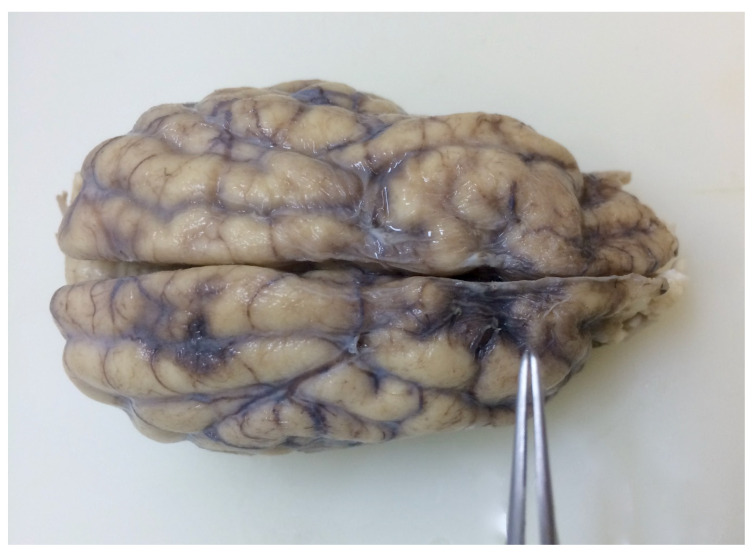
Dorsal view of the brain from a young Irish Wolfhound dog with bilateral cerebrocortical atrophy. There is atrophy and irregularity of gyri and enlargement of sulci.

**Figure 3 animals-11-00143-f003:**
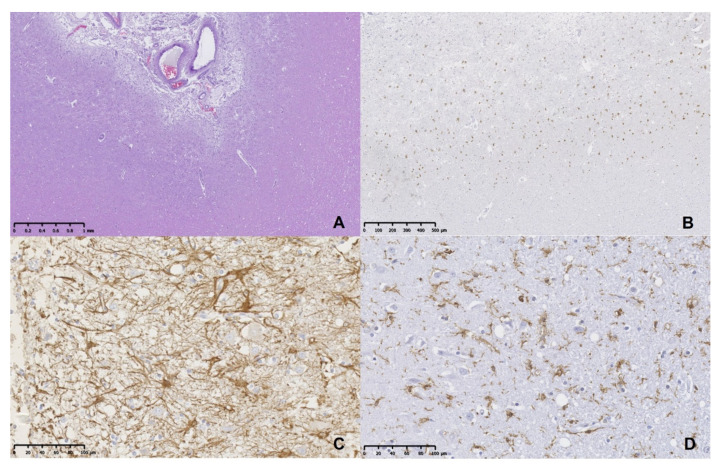
Brain histopathology and immunohistochemistry from a young Irish Wolfhound dog with bilateral cerebrocortical atrophy. (**A**). There is distension and diffuse cell infiltration of the subarachnoid space and superficial areas of pallor affecting upper cortical laminae with loss of the grey matter. Parietal cortex, HE. (**B**). Note the loss of neuronal bodies and cortical architecture disorganization. NeuN immunohistochemistry. Hematoxylin counterstain. (**C**). Numerous reactive and hypertrophied astrocytes are noted in the affected neocortical areas. GFAP immunohistochemistry. Hematoxylin counterstain. (**D**). There are abundant reactive microglia cells in the injured cortex. Iba-1 immunohistochemistry. Hematoxylin counterstain.

**Table 1 animals-11-00143-t001:** Immunohistochemical panel of antibodies used in this study.

Antibody	Reference	Manufacter	Dilution
NeuN (Clone A60)	MAB 377	Merck, Germany	1:100
Iba-1	ab5076	Abcan, Cambridge, UK	1:300
GFAP	Z0334	Dako, Denmark	1:5000
CD3	A0452	Dako, Denmark	1:100
CD20 (Clone MS4A1)	pa5-32313	Thermo Fisher Scientific, USA	1:300

## Data Availability

Not applicable.

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
