# Peer review of "Glial Response and Neuroinflammation in Cerebrocortical Atrophy in a Young Irish Wolfhound Dog"

_animals, 2021, doi:10.3390/ani11010143_

Round 1
Reviewer 1 Report
The authors describe a case of cerebrocortical atrophy in a young dog.
They describe the clinical picture and their findings in ancillary tests. They also describe the macroscopic and the microscopic lesions found in the brain and study the inflammation.
Although the study is well developed and structured, there are a few questions that should be addressed before publication.
SIMPLE SUMMARY
Line 16
Replace “on” by “in”.
Line 17
Replace “a Irish” by “an Irish”.
Line 18
Are you completely sure that this is a hereditary condition or it is just a presumption? Would it be idiopathic or probable/possible hereditary condition a better term? Did you tested for lead poisoning, thiamine deficiency or other possible causes of neuronal necrosis and loss?
Line 19
Replace “loose” by “loss”.
In line 20-21 you mention that you have described how the neuroinflammation and glial reaction have exacerbated the damage to the nervous tissue, however, this is vaguely referred in the conclusion and almost not addressed in the discussion. Could you develop this more?
INTRODUCTION
Line 37
Delete “of”, should read “mainly neurons”.
Line 39
Add “to” after seems, should read “seems to occur”.
Line 47
I would replace “has been implicated” by “has been suggested” or “has been indicated”.
Line 52
Delete “now”.
Line 52
Shouldn´t it read “it is influenced by” instead of “its influence by”?
Line 51-54 reads too long … I would separate phrases. For example: “Neuroinflammation was previously understood as a local tissue response with few or no involvement of the peripheral immune system. Nevertheless, recent data support that it is influenced by….
Line 55
I would replace “of” by “in”; “findings in one case of”.
Line 56
Add a comma before “emphasis”.
Line 57
Add a comma before “which”.
DESCRIPTION OF THE CASE
Line 61
Delete “up”.
Line 66
Either you say “indicative of a diffuse primary bilateral cortico-thalamic lesion” or “indicative of diffuse primary bilateral cortico-thalamic lesions”.
Line 74
Replace “cisterna cerebellomedullar” by “cerebellomedullary cistern”.
Line 74
How do you explain that the animal had neutrophilic pleocytosis in the CSF? Did you see meningitis? What is the relation with the condition you describe in the paper? Would it be possible to have a secondary infectious complication or you think it is part of the syndrome?
Line 78
Replace “to histopathology” by “for histopathological examination”.
Line 79
Add “neutral” before buffered.
Line 79
Replace “transversal” by “transverse”.
Line 84
Replace “by” by “with”.
Table 1
In the CD20 part, add a bracket before “clone”.
Line 97
Add “in” between “reduction and neocortical”.
Regarding gross changes, do you have a photo you could include in the publication? The lesions you describe seem to be pretty spectacular and the readers may benefit from these images in case they confront to a similar case.
Line 108
Didn´t you see any neutrophil?
Line 118
Replace “cytoplasmatic” by “cytoplasmic”.
Line 119
Replace “was” by “were”.
Line 120-121
You only detected few perivascular B and T cells but in line 108 you describe mild to moderate infiltration of lymphocytes, how do you explain this apparent discordance?
Regarding figures. It is almost impossible to appreciate the changes you describe in figures 1A and 1B. Could you please include a higher magnification?
Line 127
Replace “loose” by “the loss”.
DISCUSSION
You describe in your results the proliferation and activation of astrocytes and microglia. As it seems to be the main objective of your study, could you describe here a bit the role you think these cells may play in the disease?
Line 138
Replace “and” by “that”.
Line 145
You start the paragraph talking about “this condition” but at the end of the previous paragraph you are talking about other diseases. I would again include at the beginning of this paragraph the name of the condition you´re talking about.
Line 151
Be consistent with the way you write brain stem (line 100).
Line 157
Ruled out or made these potential causes less probable? There could be other possible infectious/non infectious diseases you did not check for …
CONCLUSIONS
I would reorganise and rephrase the information you give here.
You are also including some results and discussing them. This is not the aim of the conclusion portion of the manuscript.
Line 169
Again you mention here that there is a poor lymphocyte response but in the results you indicate that there are even moderate lymphocytic infiltrates. Please amend or explain this apparent discordance.
Line 171
Replace “brain-blood barrier” by “blood-brain barrier”.
Line 171
Wouldn´t the neutrophilic pleocytosis in the CSF mean that the blood-brain barrier is affected?
Line 171-173
I don´t understand what you mean in this phrase. Do you mean that the glial cells are activated as a mechanism of repair but they are detrimental for the degenerative and necrotic process? Could you rephrase it?
Line 175
Replace “to further the knowledge regarding its” by “to further characterize its”.
Line 175
I would replace “the knowledge that” by “The fact that” or “Knowing the fact that”.
Author Response
All comments by the reviewers have been addressed and considered, and all suggested changes and corrections (highlighted in red) have been incorporated into the revised version of the manuscript. Please find below a detailed list of responses to each of the reviewer’s comments (our answers are in italics).

Reviewer 2 Report
The authors present an interesting case report about a dog suffering a rare hereditary neurodegenerative disease. They indicate a detailed description of symptoms, biochemical and hematological patterns, RMN, cerebrospinal fluid analysis, etc. They also show what occurs at the histopathological level.
Minor revisions
Line 62: could define BUN, ALT and AF, since it does not appear more in the text.
Line 67-68: add acronyms to be used from now on; Magnetic resonance imaging (MRI).
Line 70: Define FLAIR sequences.
The cerebellum is a structure that many papers related to neurodegenerative disease. Since you did not find changes in the cerebellum, can you venture to discuss why you think there are no changes in this case? You could even compare with cases in other species, including humans.
Also, I miss a depth discussion of what occurs at the glial level and in terms of neuroinflammation. If possible, correlate the results observed in figure 1C and 1D of astrocytes and microglia with literature about these structures in neurodegenerative disease.
Author Response

(The authors gave the same response as above.)

Reviewer 3 Report
In their work, the authors describe a case of a Irish Wolfhound dog suffering a neurodegenerative disease showing a wide cerebral cortical damage with loose of neuronal bodies in a bilateral and symmetrical pattern. The authors have studied the glial components and conclude that neuroinflammation may have exacerbated the nervous tissue damage.
I think that the case report is very interesting and may be useful to enrich the scientific literature in veterinary neuropathology.
However, I have few suggestions that I hope will help to further improve the manuscript.
line 17-18: I would not state here that the disease is hereditary: later in the discussion, the authors suggest that there are no supporting data that may confirm this hypothesis.
I would additionally perform an immunohistochemistry for MHC II antibody to further demonstrate an inflammatory-related microglia activation. As the authors suggest, activated microglia has a characteristic deramified shape but it has been described (De Biase et al., 2020) that MHC II expression is enhanced in chronic inflammation.
It would be very helpful and nice (if possible) to also have information about cytokine expression in the examined brain (by IHC or PCR).
I would add the MRI images, even as a supplemental figure.
Finally, I suggest a very mild text editing.
Author Response

(The authors gave the same response as above.)
